# AR-based UI for improving IO setup in robot deployment process

Oliver Beyer Lauritsen
Aalborg University
Aalborg, Denmark
olauri18@student.aau.dk

Jonas Andersen
Aalborg University
Aalborg, Denmark
jan18@student.aau.dk

Krzysztof Zieliński
University of Southern Denmark, Universal Robots
Odense, Denmark
krzi@mmmi.sdu.dk,universal-robots.com

Mikkel Baun Kjærgaard
University of Southern Denmark
Odense, Denmark
mbkj@mmmi.sdu.dk

## ABSTRACT

Deployment of industrial collaborative robots in production lines requires expert skills and knowledge to successfully automate a workcell. For example, to set up wired connections between the machine and the robot, an electrical technician is needed, due to the lack of standardized interfaces within the industry. This paper proposes a user-friendly and efficient method for connecting wires from the device to the correct inputs/outputs (IO) of the robot. To that extent, an Augmented Reality (AR) application for mobile devices (smartphones and tablets) is developed. It uses AR technology to display a virtual representation of contextually-aware instructions to allow the user to easily connect the wires to the correct IO ports. Through user testing, it is shown that the suggested approach has a 25% lower mental demand (based on the RTLX method) and a 15% higher usability score (based on the SUS method) compared to traditional methods.

## CCS CONCEPTS

• **Human-centered computing** → **Visualization design and evaluation methods**; *Mobile devices*; • **Computer systems organization** → **Robotics**.

## KEYWORDS

Augmented Reality, Robot Deployment, User Interface Study

**ACM Reference Format:**
Oliver Beyer Lauritsen, Jonas Andersen, Krzysztof Zieliński, and Mikkel Baun Kjærgaard. 2023. AR-based UI for improving IO setup in robot deployment process. In *VAM-HRI '23*. Stockholm, Sweden, 9 pages.

## 1 INTRODUCTION

Machine tending, a process of loading and unloading parts into a CNC machine, is physically exhaustive, repetitive, and time-consuming when done manually [9]. Moreover, this process requires skilled machine operators to ensure correct CNC machine parameters, used tooling, and placement of the parts to be machined.

Due to the non-ergonomic and mundane nature of the job, which also requires some technical training, companies are having difficulties finding qualified workers. According to Deloitte report [27], there will be a shortage of 78 000 machinists and machinery mechanics in the USA by 2029.

To tackle that problem, many manufacturers automate their production lines with a robot machine tending solution - the use of an industrial collaborative robot that takes over the tedious task of loading and unloading CNC machines. While this approach can free workers from physically-demanding tasks, it increases the required skills and knowledge of operators to include robot deployment - a process of preparing the robot to tend a machine. Integration of the robot with a CNC machine is traditionally complex [25] and often requires help from third-party integrators.

The robot deployment process for machine tending typically includes the following:

- Identification if a specific task can be automated,
- Setup of communication between the robot and the machine to be tended,
- Programming of the robot software to complete the specific task,
- Training the staff to know how to operate the workcell.

While all the above-mentioned elements are challenging, this paper focuses on simplifying the process of setting up wired connections between the robot and the CNC machine, aiming at reducing the expert knowledge and skills needed for the operators to successfully deploy robot solutions. The contributions of this paper are the following:

- Proposed Augmented Reality (AR)-based User Interface (UI) for connecting wires to the robot controller,
- Evaluation of metrics (novelty and usefulness) of the proposed solution,
- Testing bias of using AR-based UI manual compared to more traditional means of providing instructions - paper and digital manual.

Traditional paper manuals are established state-of-the-art in industry and consumer products. To make the manuals non-repetitive, drawings and descriptions of tasks are usually separated. This makes it hard for users to understand the task, and how to complete it [11].

Currently, there exist no manuals on how to integrate the communication between a robot and an arbitrary CNC machine for

a machine tending task. To properly set up communication between a CNC machine and a robot, the operator might require a CNC machine manual, a robot manual, and some level of electrical knowledge. Universal Robots has made an interactive video that instructs what signal the robot needs from the CNC machine and where to connect them to the robot's control box [34]. However, it does not assist in how to obtain the signal from the CNC machine. Additionally, there exist guides for how to set up the communication if both the CNC machine and the robot have been bought as part of a package solution [14].

AR is a promising technology for enhancing **Human-Robot Interaction (HRI)** for several tasks, including assembly and maintenance tasks [12]. Research has been made specifically regarding the effectiveness of showing instructions in AR and guidelines for AR design.

For receiving instructions, Mourtzis et al. [23] found that using their AR solution for maintenance improves both time usage, cost, and usability. Additionally, Hou et al. [19] found that when specifically comparing AR to manuals, AR reduced the number of errors made and significantly improved the learning curve of trainees.

## 2 RELATED WORKS

In this paper, the state-of-the-art within AR manuals is investigated.

Choi et al. [8] have developed an AR application for a **Hand-Held Device (HHD)** that uses computer vision, or photogrammetry, for anchoring the augmented components to the real world. This allowed workers to intuitively install and inspect outfitting parts without paper drawings. They found that using their AR application significantly reduced the amount of time needed for inspections compared to conventional drawing-based inspections. Furthermore, Boeing, a manufacturer of commercial airplanes, has developed an AR solution to show their technicians real-time, hands-free, interactive 3D wiring diagrams of their planes, using an **Head-Mounted Display (HMD)** [7]. Additionally, there exist commercially available AR applications for HHDs that have been created specifically for the setup of robotic solutions. However, these solutions only showcase the end result of the setup of a robotic solution [1, 28].

Findings from Agrawala et al. [2] have shown that it is better to show stepwise visual instructions than a visual representation of the end result. They found it beneficial to show the action needed to complete the instruction. Novick et al. [24] found that step-wise visual aid leads to fewer assembly errors, less mental demand, and better recovery from potential errors.

As can be seen in the mentioned works, there exist different types of AR methods, where the most commonly used methods in research are: HHD, HMD, and projector-based AR. Research has been made on which of these three is best. HMD was found to be the least attractive despite it being mobile and hands-free because it is uncomfortable to wear, has a low field of view, and is a more expensive option [16, 26]. Projector-based AR is good in many ways such as it being the best for group observation but it is also an expensive option and different lighting can affect the projection to appear too transparent [21, 30]. HHD shows promise due to the large availability of mobile devices, it is cost-effective and easy to use, despite its disadvantages of not being hands-free and requiring attention shifts [3, 10].

To develop a UI for an AR application, several concerns have to be addressed. These concerns include how to show relevant information and how to manage the shown information in a way that reduces the number of occlusions [13]. Additionally, a UI for an HHD has to be designed so that the user can interact with the UI through the touch screen or the motion of the device [22]. To determine the best ways to display the information, Gatullo et al. [13] compared the strengths and weaknesses of several methods for general-purpose use. They found that auxiliary models, signs, and text show promise. Furthermore, Baldassi et al. [5] found that AR instructions containing motion and rotation information improve the completion time of an assembly task compared to static AR and paper instructions.

The proposed system uses HHD and displays stepwise guidance, intended for non-experts when setting up a robot for a CNC machine. In section 4, the aforementioned UI design considerations are tested.

## 3 WORKFLOW OVERVIEW

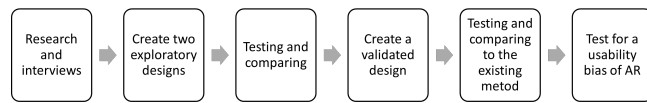

**Figure 1: Workflow for this paper**

To identify the functionalities of an AR-based manual and to provide better guidance for machine operators, an iterative design has been used and can be seen in Figure 1.

First, interviews with machine operators employed at SMEs (Small & Medium Enterprises) and with third-party system integrators (companies specializing in robot deployment in production lines) are conducted to understand the needs of the users. Based on these interviews, two UI concepts are investigated in the exploratory design phase of the project. With the received feedback from user testing, a validated design is proposed and tested against a traditional method. Lastly, a usability bias test is carried out to ensure the novelty of AR technology does not affect the obtained test results.

## 4 EXPLORATORY DESIGN

This section describes the two initial iterations of a prototype and how it was evaluated. The intention is to simplify the process of integrating communication between a robot and a CNC machine.

### 4.1 Design

Two different design concepts (Pointer UI and Animated UI, as seen in Figure 2a and Figure 2b) were developed to highlight and identify the value of the various aspects. They were made with the assumption, that all signal wires and where to connect them are known to the system. They are used to compare a concept of written instructions with a dynamic AR line and a skeuomorphic animated AR concept. The designs are based on the findings of Gattullo et al. [13] - using texts and auxiliary models, and of Agrawala et al. [2] - using step-wise instruction (a wizard application). Moreover,

Animated UI also uses the findings of Baldassi et al. [5] - using motion improves completion time.

The Pointer UI has a textbox with instructions and a simple augmentation that is a line that points from the instruction textbox to the real-world port to which the instruction is referring in the text. The Animated UI utilizes a skeuomorphic design of a wire. The augmented wire is animated and inserts itself into the real-world port to emulate the action the user has to carry out. Additionally, a textbox with instructions is visible when toggling the correct button.

The applications differ in how they show instructions to the user but share standard features such as how to navigate between the steps, how to scan the control box when starting the application (to localize itself), and how to inform the user when they are finished with all the steps. Both applications have all elements placed near the edge to reduce occlusion.

The prototypes have been implemented and tested using an iPad Pro [4]. They were implemented using Unity 2021.3.13f1 and AR Foundation 5.0.2. The application uses AR Foundation's image tracking implementation to localize itself to the control box. [31, 32]

For a robot to successfully be implemented in a machine cell, communication has to flow between the robot and the machine. This communication is done by connecting wires between the IO of the robot and the machine, where each wire represents a specific signal [34]. The task for the participants when testing was to correctly connect these signal wires to the I/O ports of a Universal Robots (UR) UR3e control box.

### 4.2 Test

The two UIs were tested against each other and a Universal Robots Academy interactive video [34], as it was the most similar medium to compare against. The video requires the user to drag the wires presented in the video into the correct ports. Both before the wires appear and between each wire, it gives the instructions and additional information using audio and a text box.

For evaluating these three solutions a within-group test with six participants was carried out.

The task was to connect four signal wires to a UR control box. All test participants used the three solutions, with the order of the solutions normalized. Before the test began each participant was asked about their experience with a control box. Afterward, they were interviewed to gain feedback. During the interview the participants were asked to rate the solutions (on a scale of 1-5) and if they agreed with statements regarding the solutions. The statements can be seen in Table 2. Table 3 shows the categorised feedback.

### 4.3 Results and further work

Several key learnings were found:

- The participants rated the interactive video the highest (Table 1). They mentioned that they rated it the highest because it was a more polished solution.
- Both AR UIs were faster to use than the interactive video, where Pointer UI was the fastest on average (Table 1).
- Majority of the respondents agreed that having the instructions on the control box was better (Table 2).

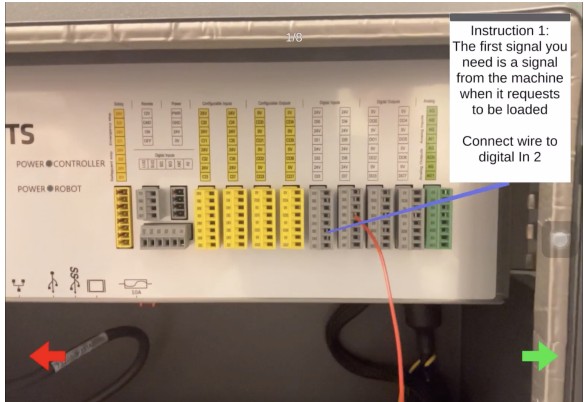

**(a) The Pointer UI**

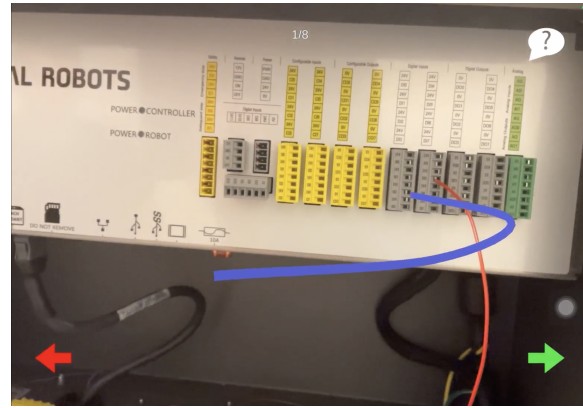

**(b) The Animated UI**

**Figure 2: Two design concepts, giving the instruction for step 1 and showing what port to insert the wire**

- Majority of the respondents agreed that animated instructions are more intuitive (Table 2).
- Majority of the respondents mentioned that they preferred to have text instructions available at all times (Table 3).
- Participants were confused as to how to connect the wires in the IO ports.

Based on the feedback in Table 3, several features were added. The following section describes the improvements.

**Table 1: The rating and time used for each solution. The rating is on a scale from 1 to 5, where 1 is worst and 5 is best.**

| Solution | Median Rating | Median time |
|---|---|---|
| Interactive Video | 4.25 | 150 s |
| Pointer UI | 3.75 | 79 s |
| Animated UI | 3 | 136 s |

## 5 VALIDATED DESIGN

This section describes the design, testing, and results of the validated design - AR Wizard Application, referred to as ARWA.

**Table 2: The participants' answers to statements regarding the solutions. Some participants neither agreed nor disagreed with some of the statements.**

| Statement | Yes | No |
|---|---|---|
| Instructions on the machine are easier to follow | 5 | 0 |
| Animated instructions are more intuitive | 3 | 2 |
| Animated instructions were occluding | 1 | 4 |
| Users are confident while using the guide | 4 | 0 |

**Table 3: The categorized feedback from the participants and how many participants mentioned it.**

| Feedback | Mentions |
|---|---|
| Want instructional text at all times | 5 |
| Labeling of wires in UI | 5 |
| Struggles to hold iPad | 4 |
| Want clearer iPad camera image | 4 |
| Animated UI imprecise | 3 |
| Want schematics of I/O wiring | 3 |
| Want use from iPad when put on the table | 2 |
| Do not know where to hold iPad | 2 |
| Want feedback from scanning | 2 |
| Make UI text resemble port symbols | 2 |
| Likes audio of interactive video | 2 |
| Want smaller device | 1 |
| Want more realistic wire animation | 1 |
| Combine all solutions | 1 |

## 5.1 Design

A validated design is based on the learnings from the exploratory design test results - user feedback in section 4. This iteration has the same functionalities as the previous two but with additional features to improve usability. Figure 3a shows the UI.

These features include the combination of the two previous UIs, where the pointer text is always shown and a toggle button is created that toggles the animated instructions. The addition of a screenshot button enables the user to take a screenshot of the AR environment which is then displayed when they place the iPad horizontally with the camera covered. These screenshots are instructions-specific and the newest screenshot is saved for each instruction. If no screenshot is saved for the current instruction, an image reminding the user to take a screenshot is shown.

Furthermore, an area designated for holding the iPad is created. Interactions with this area of the screen are ignored. This holding area is marked with the words "Hold Here" in the UI.

A basic implementation of wire tracking is implemented, to label the different wires in AR. This implementation utilizes fiducial markers to recognize and track the position of the wires. When a wire is being tracked, the signal of this wire is then shown as the text above the wire, with a line pointing to the wire. Figure 3b shows the wire tracking of multiple wires. To reduce the amount of occlusion, the label of a wire is removed after two seconds of being outside the camera view.

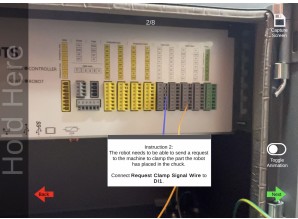
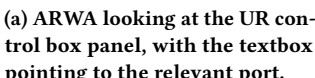

(a) ARWA looking at the UR control box panel, with the textbox pointing to the relevant port.

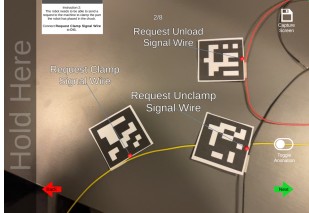

(b) Tracking of three different signal wires. ARWA shows the label of each wire, to help the user locate the correct wire.

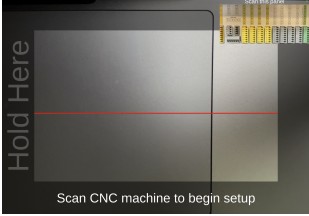

(c) ARWAs UI when it is initially searching for the control box' I/O panel.

**Figure 3: Components of the validated design**

To assist the users in connecting the wires, a video is made. This video is displayed when starting the application and is looped until the user proceeds with the setup. After the video is closed, a scanning screen is shown. This is done to make the scanning step clear to the user. The scanning screen can be seen in Figure 3c. It consists of an image of what the app is searching for, in the top right of the screen, and a moving red line to indicate that it is currently searching, inspired by a barcode reader.

For this iteration, text is added to all buttons to make the functionalities of these more unambiguous to the user [36].

## 5.2 Test

In this iteration, two tests are carried out. First, a comparison of the performance of ARWA to a control test is performed.

Akçayır et al. [3] found that the novelty effect might have an impact on the perceived usability of an AR system and that this usability bias diminishes over time, leading to skewed results in the initial test results. Therefore, a second test is created to determine the bias.

*5.2.1 ARWA test.* A control application, that does not utilize AR is created. It is designed to resemble a book and to simulate a manual, such that ARWA could be compared to a traditional method. It consists of five pages, where the two first pages show an example in images and text, of how to connect and secure the wires in the connectors. After this, there are two pages of instructions with four instructions on each page. The instructions are written instructions, where the text is the same as the text shown in ARWA. On the last page, an overview of the control box' IO ports can be seen, taken from the official UR manual [33]. To browse through the pages of the app, the back and next buttons are available at the bottom of

the screen. The next button goes to the next page, and the back button goes to the previous page.

The two mediums were compared using a between-groups test design, with nine participants in each group. This was chosen to prevent the participants from gaining experience with the task at hand and to reduce the testing duration for each participant. Before testing, each test participant was asked what they did for a living, how much experience they have with electronics, and how much experience they have working with robots. This was done to ensure the participants had little to no experience with setting up the IO of a robot and little experience with electronics.

The same test setup was used for both methods. The test setup consisted of a UR control box, a screwdriver, and eight wires with fiducials attached, where one end of the wires was taped to the table, to the right of the UR control box. This was done to simulate the wires being connected to another machine's IO, the "connected" end of the wires were each labeled as a signal wire, with the name of the signal corresponding to the ones used in the initial test and the UR interactive video. The label of the wires could not be seen from the user's position without closing the door of the UR control box, this was done to simulate them checking where the wires were "connected" to the other machine.

The task for the test participant was then to connect the other end of a signal wire to the correct IO port. Before the participants were asked to connect the wires, they were briefly explained the task and the context of the task at hand. After this introduction, they were handed the iPad with either the control or ARWA running. They were then introduced to the functionalities of the given app.

For the control, they were told to think of the iPad as if it was a book, where they could go through the pages by pressing the "Next" and "Back" buttons. They were then told what the different pages contained. For ARWA they were also introduced to all the different functionalities and were told that if the instructions seem to drift, they could hold the iPad so that it can see all the IO ports to re-localize the instruction.

While performing the task, the participants were told to ask questions if they had any and to think aloud. Furthermore, they were recorded using a GoPro camera while performing the task to analyze and compare their behavior. Several metrics were also recorded while the participants performed the task. These were:

- Total Time spent performing the task
- Time spent per step
- Number of mistakes
- Type of mistake
- Interactions with the UI (only recorded for ARWA).

The mistakes were counted by inspecting the panel after a participant was done. Each port that did not have the correct wire inserted counted as one mistake.

When the participants were done with the task, they were asked to fill out a questionnaire.

*5.2.2 Bias test.* To measure the usability bias of AR, a test was designed to compare AR to other mediums that provide visual information to a user. The different mediums were: AR, physical objects the user was not allowed to touch, and images. Figure 4 shows the test setup of the bias tests.

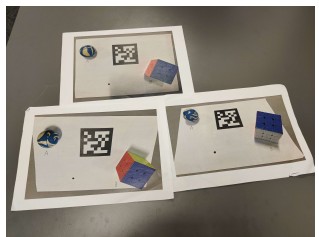

**(a) Image usability bias test setup. The pictures used for conveying visual information.**

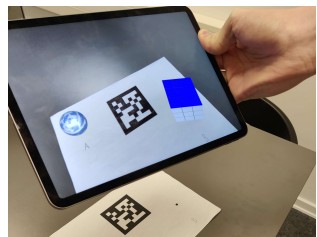

**(b) AR usability bias test setup. The iPad with the augmented reality objects placed on the paper in front of the user.**

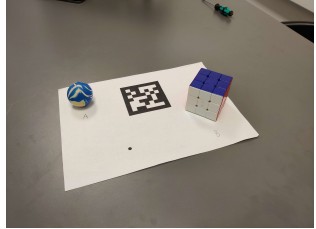

**(c) Physical usability bias test setup. Physical objects of the ball and cube are placed on the paper in front of the user.**

**Figure 4: The three different mediums used for testing user usability bias**

The 3 mediums were compared using a between-groups test, conducted on 18 participants, with 6 participants in each group.

When testing for the usability of AR technology, an A4 paper with a fiducial was placed in front of the participant. The paper had three dots, where two of the dots had the letters "A" and "B" written beneath them, respectively. The user was then handed the iPad and asked to scan the fiducial. When scanned, a small blue and white virtual ball, and a virtual Rubik's cube were shown at dots A and B, respectively. When testing with physical objects, the fiducial was placed in front of the participant, this time they were not given the iPad, but a blue and white bouncy ball was placed on dot "A" and a Rubik's cube was placed on dot "B". They were then told not to touch the objects. For testing the usability of images, three images were placed in front of the participant instead of the fiducial in the two other setups. The images were of the physical test setup, taken from 3 different angles.

All participants were tasked with answering the same questions, using the visual information given to them. The questions were heuristically designed to be equally difficult to answer independently of the given medium, such as "Can you see two objects in front of you?", "What color is object A?", and "Which object is the largest?"

## 5.3 Results

Since using AR to provide a manual for IO setup has not been explored in depth, it is important to evaluate the usability of these instructions, to determine if AR can enhance the efficiency and effectiveness of the task at hand. Furthermore, task completion time

and errors should be considered as key metrics for evaluating the solution. Previous work, regarding the development of AR systems, used **System Usability Scale (SUS)** and **Task Load Index (TLX)** methods to evaluate the usability of these systems [6, 18, 35].

The SUS method is the most widely used measure of perceived usability. There exist several variations of the method, however, the standard version is the most reasonable to use to remain comparable to the majority of previous work [20]. The obtained usability score ranges from 0-100 and can be used to compare the usability of a system to an established database. A SUS score of 68 is considered average [29].

The NASA TLX method is used to measure the task load of a system - the cost of accomplishing a task. The task load of a system is determined to negatively correlate to the usability of the system. [35] In this paper, **Raw TLX (RTLX)** method, a compressed version, but equally sensitive, is used. It consists of 6 questions, each representing a subcategory of mental load [15].

Moreover, AR is believed to have a high novelty effect, therefore to measure it, **User Experience Questionnaire (UEQ)** is used. It measures user experience (UX) on 6 scales: Attractiveness, Efficiency, Perspicuity, Dependability, Stimulation, and Novelty [17]. Only the novelty scale is measured in this paper.

Test results are analyzed using statistical tests to determine the results' significance. Normality and homogeneity of variance were tested on the sampled data to determine an appropriate test for comparing the samples. Normality was tested using Shapiro-Wilk and homogeneity of variance was tested using Bartlett's test. For comparing two samples, a t-test is used if the samples are normally distributed and the variance is homogeneous, otherwise, a Mann-Whitney-Wilcoxon test with continuity correction is used. For comparing more than two samples the ANOVA test is used for normally distributed data with homogeneous variances otherwise a Kruskal-Wallis test is used. For all tests, a significance level of 5 % is used.

*5.3.1 ARWA Results.* In Figure 5a and Figure 6a, the RTLX scores for ARWA and the control can be seen, respectively. For RTLX a lower score indicates a smaller task load, except for the performance parameter where a higher score indicates a better performance. It can be seen that the median task load of ARWA is lower on every parameter except for performance where they score the same.

**Table 4: p-values when comparing ARWA and Control on the parameters: Mental Demand (MD), Physical Demand (PD), Temporal Demand( TD), Performance (P), Effort (E), Frustration (F), and Usability (U), respectively.**

|  | MD | PD | TD | P | E | F | U |
|---|---|---|---|---|---|---|---|
| p-value | 0.173 | 0.204 | 0.541 | 0.963 | 0.313 | 0.437 | 0.655 |

The p-values from comparing the RTLX parameters can be seen in Table 4. None of the RTLX results were found to have a significant difference between the two methods, however, this is likely due to the small sample size. Mental demand and physical demand show relatively low p-values of 0.173 and 0.204 respectively, indicating that there is a trend in differences in mental and physical demand.

In Figure 5b and Figure 6b, the SUS scores of ARWA and the control can be seen. ARWA has a median score of 82.5 and the control has a median of 67.5. It can be seen that the ARWA has 15% higher usability. From Table 4, it can be seen that the results do not show a significant difference as the p-value is 0.655. Looking at the SUS database the score of ARWA corresponds to "Excellent" performance, whereas the score of the control is of "OK" performance.

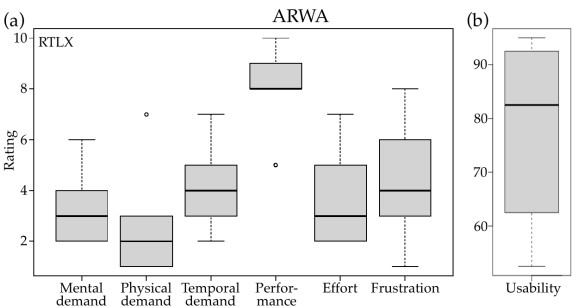

**Figure 5: Results for ARWA test: (a) Boxplot of RTLX results, (b) Boxplot of usability.**

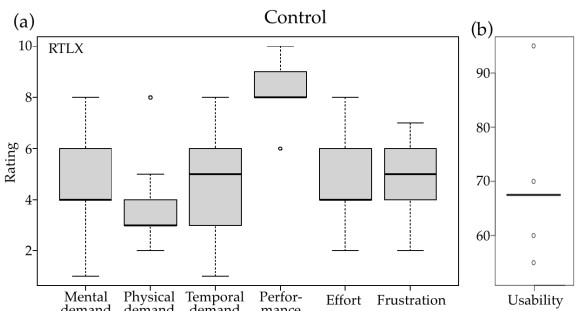

**Figure 6: Results for control test: (a) Boxplot of RTLX results, (b) Boxplot of usability.**

The average step time can be seen in Figure 7a and the average completion time can be seen in Figure 7b. The control was found to be the fastest method. The completion time and all step times, except for the first and the last are shorter.

The completion time for the control is 235 s faster than ARWA on average and it is close to being significant, having a p-value of 0.07. Step 2 in Figure 7a can be seen to have both a larger mean and a larger standard deviation, compared to the other steps. Two of the participants using ARWA had a time duration for step 2 of 390 s and 689 s. This is two and three times more time used at step 2 than the participant that used the third most time at that step. It was observed that for one of these participants, this was due to going back to correct an error. The other participant realized an error was made during step 1 and used the time in step 2 to correct it but then also made another error while correcting the first error. This resulted in them using a lot of time at step 2 to identify and correct both errors.

The number of mistakes each participant made when they stated they had completed the task, can be seen in Table 5 and Table 6. It

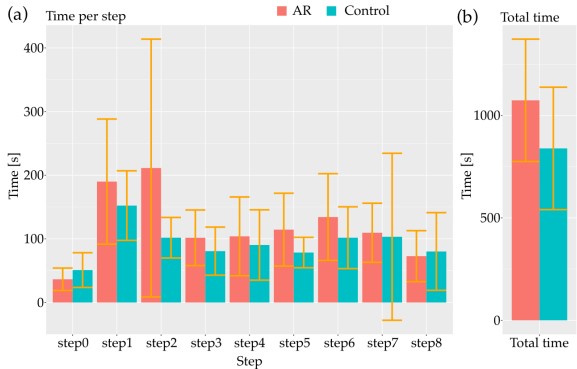

**Figure 7: Barplot of: (a) the mean time for each step grouped by the method. Step 0 is the instruction on how to insert wires and the time the participant used to scan the panel when using ARWA; (b) mean completion time. The orange line indicates the standard deviation.**

can be seen that two participants that used ARWA made mistakes and four that used the control made mistakes. The control has also twice the number of mistakes, compared to ARWA. Three different types of mistakes were observed:

- The wire was unconnected,
- The participant used the correct wire but connected it to the wrong port,
- The participant connected the wrong wire to the correct port.

The different types of mistakes were observed to have different causes. Participants using ARWA mainly made the mistake of having a wire unconnected. This was most often due to the participant not correctly securing the wire in the connector, and it, therefore, fell out after they stated that they were finished with the task. Participants using the control mainly made the mistake of confusing two wires of the same color, thereby, resulting in two mistakes.

The difference in mistakes is believed to be partially due to ARWA's wire identification. It allowed the participants to more easily identify what wire they were searching for.

**Table 5: Participant mistakes using ARWA**

| Participant | Mistakes |
|:---:|:---:|
| 1 | 0 |
| 3 | 0 |
| 5 | 2 |
| 7 | 3 |
| 9 | 0 |
| 11 | 0 |
| 13 | 0 |
| 15 | 0 |
| 17 | 0 |
| Total | 5 |

**Table 6: Participant's mistakes using the control**

| Participant | Mistakes |
|:---:|:---:|
| 2 | 0 |
| 4 | 0 |
| 6 | 4 |
| 8 | 0 |
| 10 | 0 |
| 12 | 2 |
| 14 | 3 |
| 16 | 1 |
| 18 | 0 |
| Total | 10 |

Data regarding UI interactions of ARWA can be seen in Table 7. Taking screenshots was used by all participants except one. It was mostly used to take a picture of the panel with the instruction pointing to the port for the current step. The participant that did not use the screenshot function, placed the iPad in a way so the reminder image was never shown. When the reminder image is being displayed, it occludes the instructions. This might have forced the other participants to use the screenshot function more than necessary.

It can be seen that none of the participants used all the features and some specific features were for the most part unused. The hold area was only used for a short duration by three of the participants. It is believed that either the screenshot functionality was used to circumvent this problem or having the instructions on how to connect wires made it clear, that the participants would need both hands to insert the wires and they were required to place the iPad on the table to connect the wires.

Participants did not interact with the instruction text box. The few interactions with the textbox were due to it not occluding necessary information, and if participants found the placement to be sub-optimal they would change it once at the start of the test.

Only Participant 11 enabled the 3D animation for a duration of 88 s. It suggests that text instructions were preferable or at least sufficient for this task. It also suggests that they might prefer to not have the animations at all or simply have forgotten the feature was available.

*5.3.2 Bias Results.* In Table 8, the SUS results (obtained from a questionnaire given to the users after completing the test) of the bias test can be seen. The median score for AR is higher than both physical and images mediums but the average is lower, this difference has a p-value of 0.979 which means that no significant difference in usability scores was found.

The UEQ results from the ARWA test can be seen in Table 9. The novelty factor was calculated using the data analysis tool provided by the authors of UEQ [17]. Again, no significant difference was found with a p-value of 0.499. These results indicate that the novelty factor of AR is less than expected and has no significant impact on the measured usability and task load of the system.

## 6 DISCUSSION

Initial measurements of the novelty effect of AR and ARWA seem to indicate that no significant novelty effect and related usability bias is seen regarding AR technology. These results lead to the belief that the measured usability of ARWA corresponds to the long-term usability of the system. The non-AR control is faster for most of the steps and the total time compared to the AR solution. However, participants using ARWA had half the amount of mistakes, which arguably offsets the time difference with the control being faster, as it would require additional time to correct these mistakes before a system would be operational. ARWA was rated better in both SUS and RTLX except for performance where they were rated equally. The SUS score of ARWA is classified as 'excellent' when compared to the benchmark, whereas, the SUS score of the control is classified as 'OK'. Although there was no significant difference between the SUS and RTLX ratings for the two solutions, the results suggest that ARWA has better usability and task load.

**Table 7: Recorded data for each participant using ARWA. Screenshots are the number of screenshots taken during the task. The active screenshot is the percentage of the total time when the iPad was on the table with a screenshot displayed. W/o screenshot is the percentage of total time where the iPad was on the table without a screenshot. The hold area shows the time the participant touched the hold area. Move textbox shows how many times the textbox was moved. Scale textbox is how many times the textbox was scaled in size. Animation is the percentage of time with the 3D animation active.**

| Participant | Screenshots | Active screenshot (%) | W/o screenshot (%) | Hold area (s) | Move textbox | Scale textbox | Animation (%) |
|---|---|---|---|---|---|---|---|
| 1 | 10 | 62.5 | 1.1 | 0 | 4 | 6 | 0 |
| 3 | 11 | 40.8 | 0.8 | 0 | 2 | 0 | 0 |
| 5 | 8 | 75.1 | 1.1 | 139.5 | 0 | 0 | 0 |
| 7 | 0 | 0 | 0.7 | 0 | 0 | 0 | 0 |
| 9 | 8 | 67.1 | 0 | 0 | 0 | 0 | 0 |
| 11 | 5 | 42.4 | 9.5 | 47.3 | 0 | 0 | 5.9 |
| 13 | 11 | 56.4 | 0.6 | 65.3 | 0 | 0 | 0 |
| 15 | 8 | 66.2 | 10.4 | 0 | 0 | 0 | 0 |
| 17 | 9 | 75.3 | 0.3 | 0 | 0 | 0 | 0 |

**Table 8: Median, average, and standard deviation of the mediums' usability**

| Medium | Median | Average | SD |
|---|---|---|---|
| AR | 82.50 | 80.83 | 9.83 |
| Physical | 78.75 | 82.08 | 14.09 |
| Images | 76.25 | 82.08 | 12.19 |

**Table 9: Novelty score measured by EUQ questionnaire, calculated using the data analysis tool designed for it.**

| | Novelty Score | SD |
|---|---|---|
| ARWA | 0.03 | 0.54 |
| Control | -0.14 | 0.49 |

All testing was done in a laboratory setting with university students. More testing is needed to determine the extent of the benefits of AR in a real-life setting, with the target users.

The tests were also conducted on a UR control box. The UR control box has a clean layout of the IO ports and the port names can both be seen on the individual ports and an overview of the ports can be seen inside the box. This could lead to visual guidance being less beneficial in this scenario as the user already has a lot of information directly available on the box.

It was observed that the lighting of the test setup had a significant effect on the performance of AR localization. For the iPad to reliably track the image target, there had to be good lighting conditions. This meant that in some of the tests the iPad had some trouble localizing the image target. This in combination with the drift of the internal localization of the iPad, led to an increase in time for the participants to see where the instructions pointed to.

The scaleability of using AR for instructions cannot be concluded, based on these findings. However, as ARWA did not show a significant improvement compared to the control, it requires more research to know if AR adds sufficient value as creating AR instructions takes more effort than written instructions. Due to the number of different instruction sets needed to be made, it is a concern that AR instructions will take too much effort to make.

## 7 CONCLUSION

In this paper, an AR wizard application for setting up IO between a robot and a CNC machine was created, to simplify the process and to reduce the expert knowledge and skill needed for the operators to successfully deploy robot solutions.

Exploratory design was made to validate if an AR solution to this problem is feasible and what features are needed for the user to successfully perform the task. Based on the findings of this test, further work with the wizard application was made. The validated design was then tested against a control where instructions were given to the user in written form. The two methods were tested using 18 participants - university students. The results show trends toward lower task load, higher usability, and fewer mistakes for the ARWA, though at the cost of a slower setup completion time. More data is needed to determine if these trends show significance.

It is believed that AR shows benefits compared to non-AR solutions, but that more work is needed to be done to find out how much an improvement AR is compared to the effort needed to create the instructions, or if written instructions are sufficient.

## ACKNOWLEDGMENTS

This paper was produced in collaboration with Universal Robots. This work is partially funded by Innovation Fund Denmark, as part of Industrial PhD program.

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
