# OpenReview forum: "AR-based UI for improving IO setup in robot deployment process"
_humanrobotinteraction.org/HRI/2023/Workshop/VAM-HRI — VAM-HRI 2023 Oral_

### Official Review · Program_Chairs · 2023-02-25
**Accept**

**Rating:** 7
**Confidence:** 5

**Review:**

Review 1:

This work investigates the design and development of an AR-based UI for IO setup of robots. The authors perform an exploratory design of two UI prototypes. From the feedback gathered, a final prototype was developed to test against a baseline setup. The authors find that there are benefits to using AR in this scenario, but further work needs to be done to truly understand how much.

Strengths:
Exploratory Design: Especially in the case of developing UI, it is always great to see initial studies that help inform the final design.

Weaknesses:
Visualization Modalities: While the authors did provide motivation for using an iPad as a visualization device, the paper could have been stronger if it had included both an HMD and a projector as variables. It would have been interesting to see whether, despite the HMD being uncomfortable, participants performed better using an HMD since they would have both hands and could see directions directly situated within their environment.
Task: In the task, the setup seems relatively simple. Although only one participant tried using the animated UI, it could have made more sense to use the animated UI in a more cluttered setup. Would participants have performed similarly if the wire box was extremely cluttered? With more visual clutter, I would expect the AR aspect of the tool to provide greater improvements to the process because it could help participants pinpoint where to connect a wire.

The authors will benefit from talking about their current and future work at VAM-HRI, therefore I would argue for it to be accepted.

Review 2:

This paper explored the usefulness of an augmented reality-based user interface for conducting IO wiring setup connecting a robot to a CNC machine, comparing it with more traditional instruction manual style methods. The authors first conducted exploratory research, collecting interview data from end users and testing candidate AR visualization designs against each other. They then created a final AR design using the results of their exploratory analysis, implemented it on a handheld tablet, and tested it against a baseline in a human subject experiment. The findings were mixed, suggesting that the AR interface led to longer setup times, but fewer errors, as well as being rated as more usable and requiring less workload (though many of these findings lack statistical significance).

Strengths:
- The rigor of the design process is very high. The various stages of the research (target user interviews, exploratory design testing, AR bias testing, validated design creation, and final test against a baseline) were well thought out and were clearly beneficial to the intended contribution.
- The developed AR system seems very promising for not just IO setup for robot connection to CNC machines, but IO setup in general. In fact, I might reword some of the introduction to speak more generally about the capabilities of the developed system, then narrowing into the CNC setup problem as exemplary of the larger class of task.

Weaknesses:
- I think the final experiment could benefit from running more participants, given the lack of statistically significant results. It’s clear that some of the results are trending towards significance, but the claims made in the paper really can’t be substantiated without more data. Of course, since this is a workshop, preliminary results are acceptable.
- Relatedly, I would be careful making claims that are not backed up statistically in an eventual, more complete version of this publication. In the discussion, the authors often talk about findings (this measure is higher than that measure, etc.) that could feasibly be the result of random variation given the lack of significance.
- In the related works section, the authors mention that hand-held displays (tablets) are more attractive than head-mounted displays. The papers cited for the disadvantages of HMDs are from 2018 and 2020, before the current generation of HMD headsets (namely the HoloLens 2) was released, which significantly alleviated many of the issues mentioned (user comfort, field of view). I’d say rather that HMDs and HHDs perform well on different metrics and are useful in different scenarios (they certainly have different price points). It’s possible to imagine that HMDs would lead to shorter task times in this paper’s experiment, due to the lack of context switching and the ability to use both hands to manipulate the wires.
- Some minor nitpicks: putting the TLX and SUS results in two graphs separated by condition makes them harder to compare than if they were presented side-by-side, similarly to the timing graph. Also, the description of the AR bias test is very far apart in the text from its results. Since it seems like more of an ancillary test than connected to the main experiment, it might be worthwhile to put those in the same section.

I think this work would be a good addition to the program at VAM-HRI, and recommend acceptance.

---

### Decision · Program_Chairs · 2023-03-02

Accept (Oral)